# Acute Lymphoblastic Leukemia Developing in a Patient with 46, XY Pure Gonadal Dysgenesis (Swyer Syndrome) with Malignant Gonadal Germ Cell Tumor: A Case Report and Literature Review

**Xinyue Zhang [1], Ying Zhang [1], Jinhui Wang [1], Jie Yang [1], Shuangni Yu [2], Min Yin [1], Sijian Li [1] and Jiaxin Yang [1,\*]**

[1]   Department of Obstetrics and Gynecology, Peking Union Medical College Hospital, Chinese Academy of Medical Sciences and Peking Union Medical College, National Clinical Research Center for Obstetric & Gynecologic Diseases, Beijing 100730, China

[2]   Department of Pathology, Peking Union Medical College Hospital, Chinese Academy of Medical Sciences and Peking Union Medical College, Beijing 100730, China

\*   Correspondence: yangjiaxin@pumch.cn; Tel.: +86-010-69156204

**Abstract:** A female phenotype with strip-like gonads, 46, XY pure gonadal dysgenesis (PGD) has a high tendency to develop into gonadal germ cell tumors. We described one patient with 46, XY PGD, who had a gonadal mixed germ cell tumor (GCT) and acute lymphoblastic leukemia (ALL). This is a unique case because two malignancies developed and relapsed in one person with chromosome abnormality, and the patient is the youngest reported so far. There is an association between her GCT and ALL, as the two malignancies may share a common clonal origin and the *NRAS* mutation likely plays a role in tumor genesis. We organized MDT to formulate a suitable plan of treatment. We completed the surgery and full cycles of chemotherapy for GCT and controlled ALL by chemotherapy and bone marrow transplantation. However, unfortunately, the young life finally ended following a rare transplant rejection. We concluded that ALL likely shares common clonal origin with GCT and that gene mutations may play a role in neoplasia, which requires further exploration. In the face of such complex conditions, we need to balance the treatment of both diseases to prolong survival and improve the patients' quality of life.

**Keywords:** 46, XY pure gonadal dysgenesis; gonadal germ cell tumor; acute lymphoblastic leukemia; gene mutation; multidisciplinary team

## 1. Introduction

Germ cell tumors (GCTs) primarily occur in young people. Patients with 46, XY pure gonadal dysgenesis (PGD) have a higher risk of developing gonadal tumors, relating to the presence of a Y chromosome or Y-chromosome material; gonadoblastoma is the most common one. It is a benign tumor, however, 50–60% are precursors to germ cell malignancy [1]. Malignant germ cell tumors comprise a heterogenous group divided into several histological subtypes, including dysgerminoma, yolk sac tumor (YST), immature teratoma (IMT), embryonal carcinoma (EC), non-gestational choriocarcinoma, and mixed germ cell tumor. GCTs are characterized by pelvic masses with elevated tumor markers such as alpha-fetoprotein (AFP), β-human chorionic gonadotropin (β-hCG), lactate dehydrogenase (LDH), etc. [2]. Due to sensitivity to chemotherapy, the prognosis of GCTs is usually favorable after surgery and chemotherapy.

Acute lymphoblastic leukemia (ALL) is more frequently reported in children and adolescents, most commonly originating from T- or B-lineage lymphoid progenitor cells [3]. The presenting symptoms of ALL include bruising or bleeding due to thrombocytopenia, pallor and fatigue from anemia, and infection caused by neutropenia. Leukemic infiltration

of the liver, spleen, lymph nodes, and mediastinum is common at diagnosis [4]. Chemotherapy is the first-line treatment. Targeted therapy and immunotherapy are also commonly used [3], while allogeneic hematopoietic stem cell transplantation (HSCT) is used much more commonly after relapse [4]. The five year overall survival rate has reached 90% as a result of standard treatment and therapeutic progress.

Herein, we report a rare case, in which a 13 year old patient with 46, XY PGD was diagnosed with acute lymphoblastic leukemia shortly after a gonadal mixed germ cell tumor.

## 2. Case

A 13-year-old girl visited a doctor because of her abnormal bleeding on 25 March 2018. At that time, she thought it was menarche, but she did not have regular menstruation since then. The ultrasound showed an infantile uterus and mixed masses on both sides of the adnexa on 4 November 2018. The serum AFP was 4833 ng/mL.

She underwent laparotomy exploration at a local hospital in Guangdong, China, on 26 January 2019. The intraoperative visualization showed an infantile uterus, with a 12 cm right adnexal mass and a 6 cm one on the left, without ascitic fluid and other suspicious metastatic nodes. The mass on the right was completely removed, and the left adnexa was resected. The post-operative pathological analysis revealed that the mass on the right was a mixed germ cell tumor, mainly constituted by dysgerminoma and grade two immature teratoma; the left was a germ cell tumor mixed with a sex cord-stromal tumor. The germ cell tumor was primarily dysgerminoma and a small amount of gonadoblastoma. Peripheral blood karyotype analysis showed 46, XY. The patient was diagnosed with 46, XY PGD with a mixed germ cell tumor classified as stage IC2 and received BEP chemotherapy (cisplatin 30 mg daily, days 1–5; etoposide 100 mg daily, days 1–2; bleomycin 15 mg daily, days 1–3) after surgery, beginning on 22 February 2019.

After one cycle of chemotherapy, the platelets continually decreased to $48 \times 10^9$/L, and the white blood cells gradually increased to $26.49 \times 10^9$/L. A bone marrow biopsy was performed on 21 May 2019, demonstrating acute lymphoblastic leukemia, with lymphoblasts accounting for 39%. Immunophenotyping showed primitive cells account for 49.6% of nuclear cells, which were positive for CD10, CD19, CD38, partially expressed cCD22, CD20, and HLA-DR. Bone marrow chromosome karyotype analysis showed 47, XY, +14 [10]. The leukemia fusion gene detection, immunoglobulin heavy-chain (*IGH*) rearrangement, and *TP53* were all negative, but a point mutation of *NRAS p.G12D* was identified. Thus, the diagnosis of high-risk B-cell ALL with *NRAS* gene mutation was defined.

Given the diagnosis, further chemotherapy, including VDLP, CAM and HD-MTX, was used to treat the leukemia, and the patient achieved morphologic complete remission (CR) of her ALL in August 2019. However, two weeks later, the restaging bone marrow biopsy showed relapsed disease with 14% morphologically abnormal B cells. A CT scan and an ultrasound showed severe enlargement of her spleen (19.8 × 4.4 cm). From September 2019 until January 2020, she took trametinib. The complete blood count (CBC), on 25 November 2019, showed white blood cell counts of $14.5 \times 10^9$/L and platelets of $40 \times 10^9$/L, which may have been from the hypersplenism.

The patient was referred to our hospital in December 2019. A CT scan showed a 13.2 × 8.3 × 20.5 cm mass and a 4.8 cm lesion located in the abdominal and pelvic, and serum AFP was 18,293 ng/mL, indicating relapse of the germ cell tumor. She underwent surgery on 17 December 2019. The mass, right adnexa, and spleen were resected (Figure 1). The postoperative pathological analysis revealed grade three immature teratoma and yolk sac tumor component without suspicion of a metastatic lesion; the spleen was consistent with B-cell lymphoblastic leukemia/lymphoma (Figure 2). After surgery, the patient received four cycles of BEP chemotherapy (cisplatin 50 mg daily, days 1–3; etoposide 150 mg daily, days 1–3; bleomycin 23 mg, day 1; 23.5 mg, day 2) until March 2020. Serum AFP decreased to 4.78 ng/mL after the second cycle of chemotherapy.

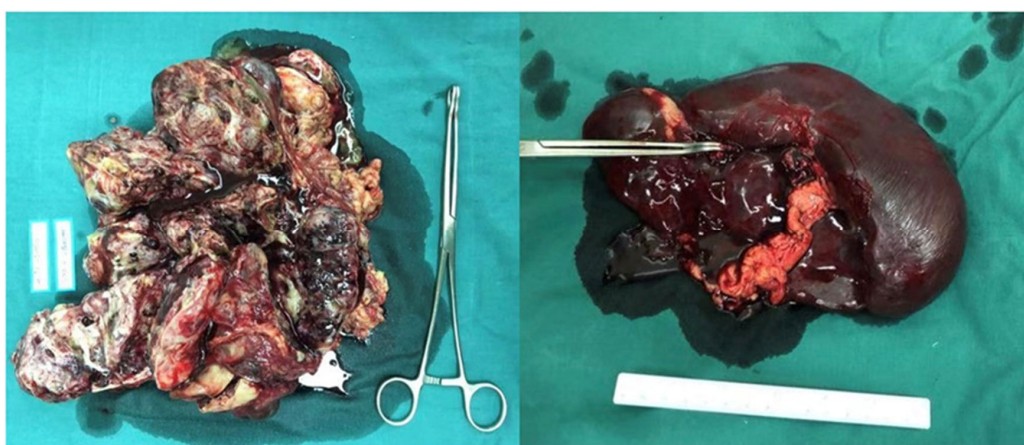

**Figure 1.** The mass located in the abdominopelvic cavity and the spleen.

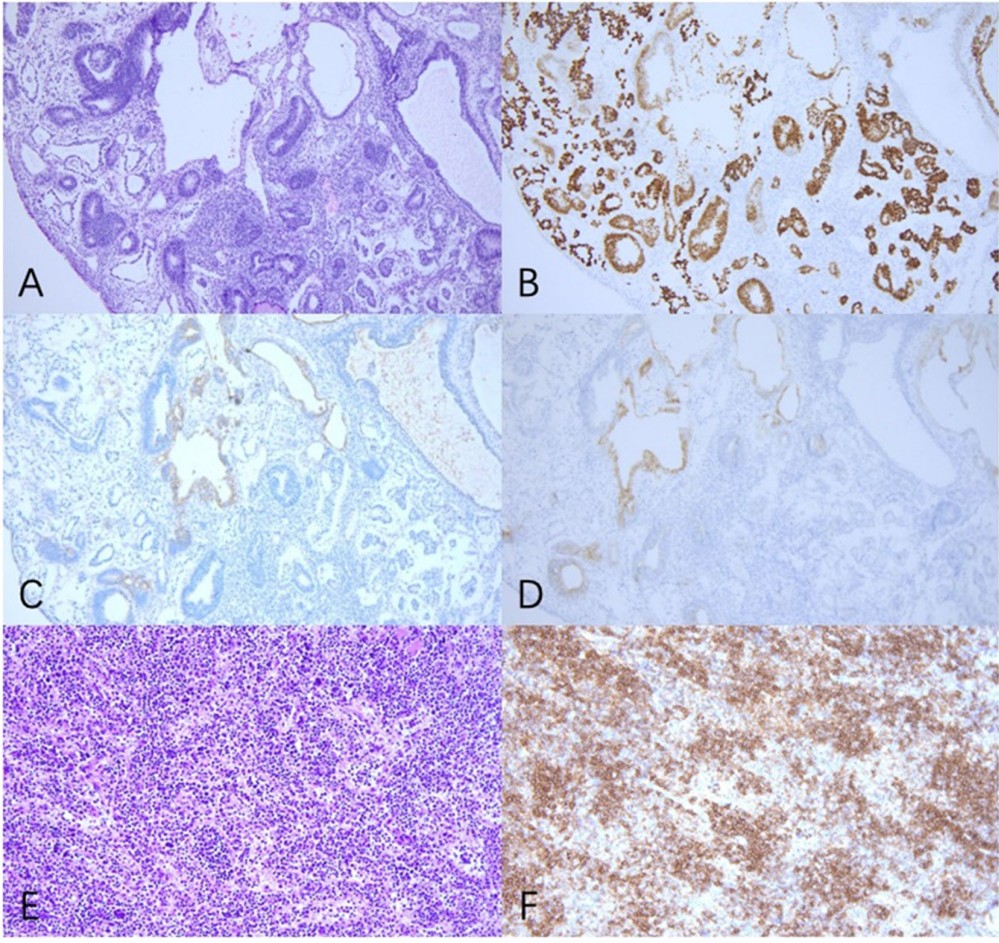

**Figure 2.** Pathology revealed that the tumor in the abdomen and pelvic cavity was IMT with YST components (**A–D**), and the spleen was B-cell lymphoblastic leukemia/lymphoma (**E,F**). (**A**) Primitive neural tube and yolk sac tumor cell (H&E 100×). (**B**) Positivity for SALL4 immunostaining in tumor cells (100×). (**C**) Positivity for AFP immunostaining in tumor cells (100×). (**D**) Positivity for GPC3 immunostaining in tumor cells (100×). (**E**) The spleen (H&E 200×). (**F**) Positivity for CD20 immunostaining in tumor cells (200×).

On 13 April 2020, a bone marrow biopsy indicated the relapse of leukemia once again. She then undertook three cycles of VICP chemotherapy (vindesine 4 mg, days 1 and 8; Idarubicin hydrochloride 10 mg, days 1 and 8; cyclophosphamide 1 g, day 1; dexamethasone

10 mg days 1–8) and one cycle of HR3 chemotherapy (cytarabine 2 g q12h, days 1–2; etoposide 100 mg q12h, days 3–5; dexamethasone 20 mg days 1–5). The bone marrow aspiration on 11 August 2020, showed immature myeloid cells of abnormal phenotype and normally developing progenitor B cells, which indicated that her ALL was remitted with some acute myeloid leukemia (AML) components. She obtained a successful bone marrow match and received allogeneic hematopoietic stem cell transplantation (HSCT) from 26 August to 30 September. Unfortunately, the young life ended from a pulmonary hemorrhage caused by a rare transplant rejection in March 2021 (Figure 3).

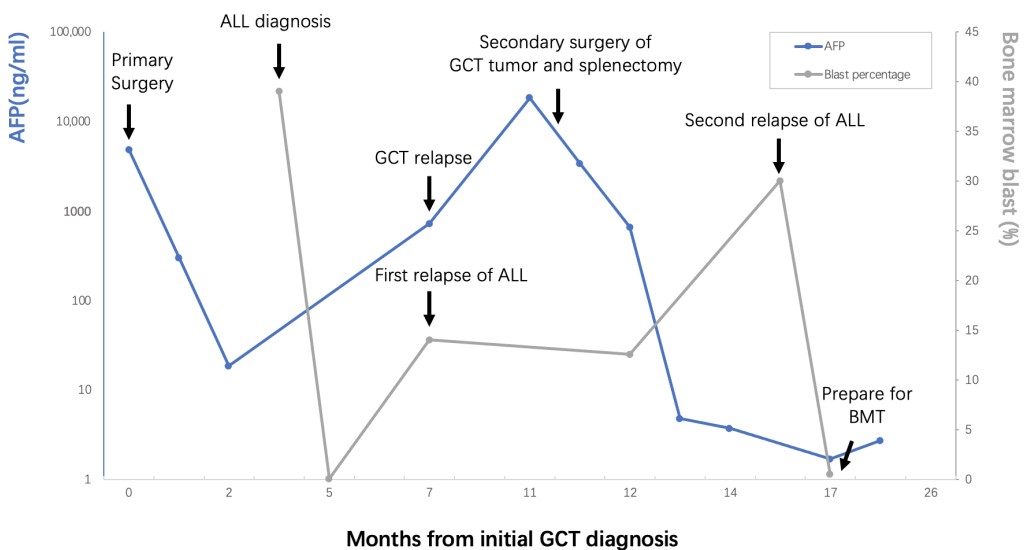

**Figure 3.** Brief clinical course of the patient's history, diagnostic process, management and outcomes.

## 3. Discussion

This is a rare and complex case as two malignancies developed and relapsed in one person with a chromosome abnormality. It is worth exploring, in this case, whether there is a link between a germ cell tumor and acute lymphoblastic leukemia, and how to deal with such a complex situation.

Sywer Syndrome (46, XY PGD) has an incidence estimated at 1/80,000. Individuals with Swyer Syndrome have female phenotype with female genital appearance and normal Mullerian structures. Testis-specific protein Y-linked (*TSPY*) gene expression, sex-determining region Y (*SRY*), Wilms tumor 1 (*WT1*), and *SRY* box 9 (*SOX9*) gene mutations are associated with tumor development [5]. Gonadoblastoma is the most common one, which is benign but can be a precursor to germ cell malignancy, most commonly dysgerminomas (DG); others include immature teratoma, embryonal carcinoma, and yolk sac tumor [1]. The prognosis is favorable when the gonadoblastoma is associated with dysgerminoma, but dissatisfactory when associated with other types. Hence, bilateral gonadectomy at the time of diagnosis was recommended [6].

We searched the PubMed database for previous literature reports of acute lymphoblastic leukemia in GCTs published in English between 1 January 1970 and 31 May 2022, using the search terms ("neoplasms, germ cell and embryonal" [MeSH] and "precursor cell lymphoblastic leukemia-lymphoma" [MeSH]). Overall, 384 citations were identified. We screened all of the abstracts and included five articles [7–11]. Then, we reviewed the references in the five articles, and three more were included [12–14]. Finally, we included eight articles; most of which are case reports. The details are summarized in Table 1.

**Table 1.** Acute lymphoblastic leukemia reported in GCTs.

| Author/Year of Publication | Case | Age | Histologic Type | Stage | Time | Reason of All | Outcome |
|---|---|---|---|---|---|---|---|
| Johnson et al. 1980 [13] | 1 | 23 | Mixed GCT (SE, EC, MT, IT) | - | 11 m | GCT | Died from infection |
| Larsen et al. 1984 [10] | 1 | 16 | GCT | - | 5 m | GCT | Died from infection |
| Bokemeyer et al. 1992 [8] | 1 | 42 | SE | IIIB | 16 m | Chemotherapy | Died from infection |
| Downie et al. 1994 [9] | 1 | 26 | YST | - | 5 m | GCT | Died from intracranial hemorrhage |
| Andersen et al. 2001 [7] | 1 | 25 | SE | | 18 m | Chemotherapy | - |
| Howard et al. 2008 [12] | 10 | - | 6 SE, 4 non-SE | - | >12 m | radiotherapy and chemotherapy | - |
| Okamura et al. 2010 [11] | 1 | 42 | SE | I | 0 m | GCT | Received CBT and alive |
| Newton et al. 2019 [14] | 1 | 14 | DG, YST | IIIA | 24 m | chemotherapy | Died from ALL |

SE: seminoma; non-SE: non-seminoma; EC: embryonal carcinoma; MT: mature teratoma; YST: yolk sac tumor; DG: dysgerminoma.

A total of 17 cases of acute lymphoblastic leukemia have been reported in GCTs: 13 of which were testicular GCTs, three were mediastinal GCTs, and one was ovarian GCT. Furthermore, nine of them were seminoma, six were non-seminoma GCTs, one was mixed GCT, and one's histological type was not illustrated. The patients can be divided into two groups, according to the cause of leukemia: four patients' ALL was associated with GCTs; the others were associated with chemotherapy or radiotherapy for GCTs. The average time interval between the diagnosis of leukemia and germ cell tumors in the former was 5.3 months, while that in the latter was >13.7 months.

Given the temporal relationship between the two malignancies, a common clonal origin was suspected. In 1991, Kaplan et al. [15] reported one patient with the karyotype of 46, XY/47, XY+8 who developed acute myeloblastic leukemia (AML) only 7.5 weeks after diagnosis of an ovarian mixed germ cell tumor. They suspected that the germ cell tumors and the malignant transformation of hematopoietic cells might be related as the primordial cell of both are derived from the yolk sac. The leukemic cells may originate within the germ cell tumor as one of the malignant cell lines derived from malignant pluripotent stem cells. There is evidence that teratoma cells can differentiate into erythrocytes, granulocytes, macrophages, and megakaryocytes in vitro [16,17]. Chromosome 12p gains are the most common copy number alteration in GCTs, and there have been reports of hematologic neoplasia associated with germ cell tumors presenting an isochromosome gain for the short arm of chromosome 12 (i12p), which suggests that they arise from the same progenitor cell [9]. Furthermore, gene mutation seems to play a role in neoplasia. Leonard et al. reported a 23 year old male with a mediastinal germ cell tumor and AML; the malignancies were identified as *NRAS* and *TP53* somatic mutations [18]. Unfortunately, our patient did not receive chromosomal or genetic testing of the tumor tissue or blood at the time of the primary surgery, so we did not know the germ cell tumor's genetic status. However, her peripheral blood and bone marrow genetic test both showed the *NRAS* mutation, one of the most frequent mutation points of germ cell tumors [19]; in combination with the short interval time of two malignancies, we speculate an association between her GCT and ALL. Her hematological tumor cells may have originated from pluripotent stem cells of GCT; the *NRAS* mutation may be proof of tumor homology and likely participates in neoplasia. Further exploration is needed to confirm our hypothesis.

The first-line chemotherapy regimen of GCTs is the combination therapy of cisplatin, etoposide, and bleomycin (BEP) [2]. Both cisplatin and etoposide can cause leukemia. Travis et al. [20] showed that secondary leukemia developed at an average of 6.8 years (median, 5.0 years) after primary cancer diagnosis, with 25% occurring after one decade (maximum latency, 17.3 years). The risk of leukemia was accumulated dose-related. When the cumulative dose of cisplatin exceeds 650 mg (or etoposide >2000 mg/m$^2$), the risk of leukemia increases, and acute myeloid leukemia is the most common [20,21]. Several investigators found that secondary leukemia was associated with translocations of the mixed lineage leukemia (MLL) gene at human chromosomal band 11q23; the most common kind was t (4,11) [7,8]. In this case, the latency time between GCT and ALL was only four

months, and chromosomal analysis showed no characteristic changes, so chemotherapy is not the cause of ALL.

Regardless of the cause of leukemia, when germ cell tumors and hematological malignancies exist simultaneously, the prognosis is not optimistic. In our literature review, among the six cases showing outcome, only one patient survived after receiving myeloablative cord blood transplantation (CBT) [11]; the other five patients all died of complications secondary to leukemia, such as infection and bleeding [8–10,13,14]. Chapman et al. [22] summarized the published cases of hematological malignancies in post-germ cell tumors, and the median time of survival from GCT diagnosis was five weeks, while the maximum was 44 weeks. Recent case reports with allogeneic transplantation and/or novel targeted therapies have had better survival rates than previous case reports; however, none so far have achieved >1 year [22]. Our patient was diagnosed with ALL during GCT chemotherapy and suffered GCT relapse after ALL remission. Facing such a complicated condition, we organized a multidisciplinary discussion to formulate a treatment plan accounting for both diseases. We first targeted the recurrence of GCT. Tumor cytoreduction surgery and splenectomy were performed simultaneously to decrease the tumor burden and relieve hypersplenism. Platelets returned to normal after surgery, so we won the time of adjuvant chemotherapy. However, the patient could not come to our hospital for chemotherapy during the COVID-19 pandemic, so we formulated a detailed chemotherapy plan for her. The patient completed four cycles of standardized BEP chemotherapy in a local hospital and reached complete remission (CR). After leukemia recurrence, standardized chemotherapy treatment was given by a hematologist, and bone marrow transplantation (BMT) was completed as soon as possible. Finally, her overall survival time reached 26 months. We considered the treatment effective and relatively successful.

## 4. Conclusions

Early diagnosis is essential for 46, XY PGD because of the risk of gonadal malignancy and the need for allowing early treatment. Once diagnosed, bilateral gonadectomy is recommended. ALL most likely shares a common clonal origin with GCT, and gene mutations may play a role in neoplasia, which requires further exploration. In the face of such rare diseases and complex conditions, we need to balance the treatment of both diseases to prolong survival and improve the patients' quality of life. Multidisciplinary teams can help formulate a reasonable treatment plan.

**Author Contributions:** Conception and design: X.Z., Y.Z., J.Y. (Jie Yang) and J.Y. (Jiaxin Yang); case diagnosis: J.Y. (Jiaxin Yang), Y.Z., J.W., J.Y. (Jie Yang) and S.Y.; literature review: X.Z., M.Y. and S.L.; manuscript prep-aration: X.Z., Y.Z., J.W., J.Y. (Jie Yang) and J.Y. (Jiaxin Yang); final approval of manuscript: all authors; Accountable for all aspects of the work: all authors. All authors have read and agreed to the published version of the manuscript.

**Funding:** This research received no external funding.

**Institutional Review Board Statement:** The article was exempted by the Ethical Review Committee (ERC) of the Peking Union Medical College Hospital (protocol code JS-1747).

**Informed Consent Statement:** Informed consent was acquired from the patient.

**Acknowledgments:** We thank Song Xue at the hematological department of Aerospace Center Hospital for the ALL treatment.

**Conflicts of Interest:** The authors have declared no conflict of interest.

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
