# Peer review of "Acute Lymphoblastic Leukemia Developing in a Patient with 46, XY Pure Gonadal Dysgenesis (Swyer Syndrome) with Malignant Gonadal Germ Cell Tumor: A Case Report and Literature Review"

_curroncol, doi:10.3390/curroncol29120766_

Round 1
Reviewer 1 Report
It is a well-written case report
Need language editing
Author Response
Thank you for your advice. Our manuscript does need to be reviewed for sentences and grammers, we will revise it with focus on grammers and writing style.
Reviewer 2 Report
I have no comments for authors
Author Response
Thank you for reviewing our manuscript entitled “Acute lymphoblastic leukemia developing in a patient with 46, XY pure gonadal dysgenesis (Swyer syndrome) with malignant gonadal germ cell tumor: a case report and literature review”.
Reviewer 3 Report
Thanks for submitting this case report to the journal. this is indeed a rare however not novel association of three disorders, 46 XY gonadal dysgenesis and malignant germ cell tumor and acute lymphobllastic leukemia. The association between Swyer and GCT is obvious. The association between ALL and GCT appears more complex. however it is well established that any kind of malignancy can arise as malignant transformation within malignant GCTs, i.e. the ALL is presumably of GCT origin. In my understanding of the manuscript, this hypothesis is not pointed out as clearly as it should be. Moreover, I would not support the hypothesis that RAS mutation could be the common confounding factor. RAS mutations are often aquired along multistep malignant transformation and are rarely causative resp. first steps of malignant transformation, especially for bot GCTs and ALL. In conclusion, I wdo not share the biological perspective on the discussion and I regret that I do not recognize the scientific originality that would merit publication of this paper
Author Response
Thank you for reviewing our manuscript. Your opinions are very helpful. Indeed, the cause of leukemia after the diagnosis of GCT in this patient is confusing. We reviewed the literature again and adjust this part.(lines 578-763 in revision manuscript) Previous literature has suggested that leukemia cells may be derived from pluripotent stem cells of germ cell tumors. There is evidence that teratoma cells can differentiate into erythrocytes, granulocytes, macrophages, and megakaryocytes in vitro.(lines 587-589 in revision manuscript) In this case, the patient's blood genetic test showed NRAS mutation, which is also a common mutation site for germ cell tumors. So we suspected her GCT and ALL share the same single gene mutation, the NRAS mutation may be a proof of tumor homology. As you mentioned, RAS mutations are often aquired along multistep malignant transformation. So the NRAS mutation may not be the cause of two malignacies but probably participate in neoplasia. However, due to lack of genetic test results of the patient's during primary surgery, we lack direct evidence to confirm our hypothesis. Further exploration and research are ongoing. This case also reminds us that the importance of preservation surgical specimens and genetic testing in primary treatment during our clinical work. Besides, the significance of this case lies not only in the mechanism of tumorgenesis, but also in its handling. Multidisciplinary consultation do helps us to solve such complex clinical problems.
Round 2
Reviewer 3 Report
Thank you for carefully revising the manuscript. The discussion and interpretation of the findings have been improved substantially.